# A Time-Motion and Error Analysis of Speed Climbing in the 2019 IFSC Speed Climbing World Cup Final Rounds

**DOI:** 10.3390/ijerph19106003

**Published:** 2022-05-15

**Authors:** Ruizhi Chen, Ziyuan Liu, Yuan Li, Jingke Gao

**Affiliations:** 1School of Physical Education, China University of Geosciences, Wuhan 430000, China; chenrz19@cug.edu.cn; 2School of Economics and Management, China University of Geosciences, Wuhan 430000, China; liuziyuan@cug.edu.cn (Z.L.); gaojingke6@163.com (J.G.)

**Keywords:** sport climbing, elite athlete, competition, strategy, training

## Abstract

Speed climbing has become an Olympic event. However, there have been limited studies on the athletic performance of elite speed climbers under the current IFSC rule. Thus, this study aims to perform a statistical analysis of the performance of elite speed climbers and compare the different sex of the 2019 IFSC Speed Climbing World Cup. The 384 times climbing result in total climbing time, the time of four phases, and the start reaction time were calculated. In addition, the statistical data of men and women’s total error rate in the final round, the error rate in each final round, as well as the body position and the phase when errors occurred were gathered. Several results were not found in previous studies. Firstly, there is no statistical significance between starting reaction and climbing time of male and female. Secondly, there was no significant correlation between phases of the route for male athletes. While there was a statistical correlation between adjacent stages for women, the time of women in each stage was significantly correlated with the previous stage (*p* < 0.05). The error rate of both men and women in the medal competition stage reached a high rate of ~50%. While the error rate of men in each phase of route has no significant difference, While the error rate of women in the fourth phase was significantly different from the first three parts (*p* < 0.05), gender-specific training procedures should be developed for elite athletes. Future research should test the psychological state and pressure of speed athletes in the competition.

## 1. Introduction

Speed climbing has become a sub-discipline of combination sport climbing disciplines in the 2020 Tokyo Olympic Games and an individual discipline in the 2024 French Olympic Games. The current world records (as of the submission of this paper) are 5.20 s for men and 6.84 s for women (International Federation of Sports Climbing, 2022).

Regarding the study of sport climbing, it mainly focuses on the boulder and lead. In contrast, research on speed climbing is limited. Current research on speed climbing has made some progress in the physiological characteristics of speed climbers.

With respect to the physiological demands of speed climbing, the weight of speed climbers was slightly higher than that of boulder and lead climbers. In terms of body mass index, international elite male climbers ranged from 20.1 to 22.1 and women from 19.3 to 22.4. Speed climbers weigh more than other individual athletes due to higher levels of tissue activity [1], resulting in higher skeletal muscle mass, which typically accounts for 43% of the bodyweight of an average adult male [2]. There have been relevant tests and studies on individual sport climbing athletes’ maximum blood lactic acid after a competition. After the race, the maximum lactic acid concentration cannot accurately reflect the anaerobic capacity of speed climbers. Moreover, in competitive climbing, anaerobic working ability requirements are different for different disciplines [3]. Thus, speed climbing is a typically short time racing event, and the energy metabolism of the athletes is mainly phosphocreatine and ATP; the explosive power of the athletes is high, so the speed climbing athletes tend to have higher muscle content to meet the requirements of the competition.

It is worth noting that, for specific sports, the analysis of sports performance in competitive competition can provide an essential basis for the goal achieved in training [4,5], and this has been confirmed in other sports by scientific research [6,7,8]. Although some researchers have investigated characteristics of elite athletes in lead and boulder [9,10,11,12,13,14] by competitive performance analysis, helpful suggestions are provided. However, to the best of the author’s knowledge, in speed climbing, especially considering the latest IFSC (International Federation of Sport Climbing) rules [15], there was no quantitative analysis of athletic or notational analysis of elite athletes’ performance.

Therefore, the aim of this study was the primary purpose of this paper is to quantify the performance made by elite speed climbers in terms of time-motion, the correlation between phases, and the feature of errors. The second is to compare the results between genders. Furthermore, using the findings of the current study and complementary information on the internal demands of this sport, both trainers and climbers could design specific test protocols and targeted training exercises.

## 2. Materials and Methods

### 2.1. Subjects

Considering that sport climbing is the first time in the Olympic Games, the world’s most excellent speed climbing elites attended the Olympic preliminary contests, and the finals represent the highest level of this game. Therefore, we used all six 2019 IFSC Speed Climbing World Cup as the subjects of this study. A total of 384 climbs were performed in the 2019 IFSC Speed Climbing competition finals. Access to the video is available from the China Mountaineering and Outdoor Sports Institute, so written consent was not required.

### 2.2. Variables

According to the IFSC rules, the time an athlete first touches the finish pad is their finish, so we also employ the time a hand touches or reaches the measurement height of key holds as the basis for time analysis. The start reaction time can be obtained from the official timing system. The time and error indicators of all speeds in the final rounds were sorted, including total time of climbing, time at different phases, error rate, start reaction time, the number of errors that occur, errors occur position, and errors occur phase.

### 2.3. Procedures

The study divides the speed route based on suggestions from athletes and coaches in China’s national team. We employed five locomotor phases to analyze the climbing, i.e., (P1) from the start to the end of the first acceleration phase (handhold 05), in which the movement in the first two steps starts at a speed of 0, (P2). From hold 05 to the end of the first dyno (handhold 10), (P3) From handhold 10 to the handhold 14, and (P4) From handhold 14 to the top device. 

The video clips in this study were obtained through the independent parsing of a former national team video analyst and an elite athlete with the best race time of 5.3 s. The shooting materials clearly showed all the 384 times climbing, with a video quality of 100 fps, which clearly recorded the principal movement mode of the athletes at different times and different back visual angles on the route. The selection of each critical hold time was based on the frame when the athlete’s hand touched a practical point of that key hold. After an order has been issued, the athlete’s first movement and the starting light have been triggered was defined as frame 0. One frame represented 0.02 s in this process, and the number of frames was converted to time. Based on these data, the time used by the athletes in each phase was calculated. However, not all climbing was successful. Therefore, adequate climbing time was defined as when the athlete did not make any errors and won due to opponents’ false starts. Moreover, the error indicator is relatively easy to distinguish between two experienced speed climbers. The different results generated in the statistics were observed by two raters again and finally reached a consensus on the judgment of all data. 

### 2.4. Statistical Analyses

In this study, the statistical analyses were performed using IBM SPSS for Windows, version 25. The data are presented as means ± SD. The Shapiro–Wilk test evaluated whether the variables conform to the normal distribution. Pearson correlation analysis (variables conform to the normal distribution) was used to verify the relationship between the time in different phases when the data were normally distributed, which applies to the variable gender and phase; The level of significance was set at *p* < 0.05 for each procedure. Moreover, we calculated the athlete’s error rate in the final stage, the phase, and body position when the athlete made errors, and the analysis is mainly descriptive. The chi-square test is used to examine the difference in the error rate of different genders at different stages and the difference in the error rate of different genders at different stages. If the expected value is greater than or equal to 5, the Bonferroni correction was used and pairwise comparisons were made between groups.

## 3. Results

### 3.1. Time-Motion Analysis and Correlations between Different Phases

Table 1 shows the time-motion statistics and correlation between different phases. The men’s starting time was 0.20 ± 0.05 (s). The women’s starting reaction time was 0.22 ± 0.06 (s). At each phase, the females consumed an average of 0.4 s longer than the male, and the overall time consumed was an average of 1.8 s more than that of the male athletes. The Pearson rank correlation test results showed there had no statistical significance between the start reaction time and overall time (*p* > 0.05). 

Table 2 and Table 3 show the correlation of climb times between different phases. In males, there only was a significantly weak negative correlation (*p* = 0.043; r = 0.593) between P3 and P4; the time between other phases had no statistical significance. In females, P1 showed a significantly high positive correlation between P2 (*p* < 0.001; r = 0.562), and P2 showed a significantly high positive correlation between P3 (*p* < 0.001; r = 0.733).

### 3.2. The Error Rate in Different Final Stages

There are five final stages in the whole final round, among which the small final is the competition for third and fourth places, and the big final is the competition for the first and second places. There is a significant difference in the error rate between men and women in the final stage (*p* = 0.019), Table 4 shows the stages with the highest error rates for both men and women were medal rounds, at more than 50%, and the semi-finals had the lowest, at 28.57% for both men and women. The quarter-finals differed significantly, with the men’s error rate reaching 48.21%, 2.07 times that of females.

### 3.3. The Number of Body Positions and Mechanism of Errors

The Table 5 shows that the distribution of the mechanism of errors made by male and female athletes is different. The CV showed that the variation of errors of male athletes in different positions is 49.7%, and the variation showed by female athletes is 28.66%. It can be found that the number of male athletes’ errors in the stalemate is significantly higher than that in other positions, which is 2.3-times the error in the leading position and 4-times the error in the lagging position. Furthermore, there were almost twice as many errors in the lead versus the lagging position. The women made only four errors in the lead and the trailing position, accounting for 42.11% of the errors in the stalemate position.

### 3.4. Correlation Analysis of Each Phase

Table 6 shows the number of errors made by male and female athletes in four route phases. By the chi-square test, the error rate of female athletes in the four phases is significantly different (*p* < 0.05). By the chi-square test, the error rate of female athletes in phase 4 is significantly different from that in other stages (*p* < 0.05), and there is no significant difference in the proportion of the error rate column between other stages. For men, there was no statistical difference between stages of error. In P2, which aims to accelerate, men make nearly twice as many errors as women. Women made 47.94% of the errors in the fourth period. On the whole, men have a lower error rate in the P1 and a more average error on the climbing route, while women have a more average error in the first three phases, and the possibility of error in P4 is much higher than that in other phases.

## 4. Discussion

This research provides vital information about climbing performance and the physiological requirements of speed climbing in different genders.

The main findings of the present study are that the correlation and time characteristics of different phases of the circuit for female athletes are different from those of male athletes. In males, positive high correlation was found between P3 and P4 (*p* < 0.05; r = 0.593); there was no statistical correlation between other phases. By contrast, with the female, there was a significantly positive correlation between the first three phases of the female athletes. The counterintuitive result reveals that male and female athletes should focus on different training content. Some studies have indicated that male athletes have strong upper limb strength [8,16,17], supporting technical adjustment at each phase and accelerating to the fastest speed in a few movements each phase. While climbing, women may need to maintain their upward inertia in the process of climbing due to their lack of strength. Thus, the benefits of full-route training for female athletes may be higher than that of male athletes, while the importance of explosive power and segmental route training for male athletes may be higher than that of full-route training.

Through time features of male and female athletes, we have found that speed climbing time was shorter than in previous studies [18,19]. Moreover, men were found to be more competitive in speed climbing competitions than women. The standard deviation of male athletes in the four phases is lower than that of female athletes, indicating that the level of male athletes in each phase and the total time are closer than females. 

Another essential finding of the present study was that the point of view of starting reaction differs from previous studies [2]. We found no statistical correlation between the starting reaction time and climbing time, and the starting reaction time is shorter than a previous analysis which shows that the average start reaction time of the male elite speed climbers is 0.41 (s) [2]. The equipment development and changes in the game rules may account for the difference. The official start reaction time in the 2019 IFSC competition is calculated when the athlete’s foot leaves the start device, reducing manual calculation error. In addition, according to the rules, the start command is composed of three “Di” sounds separated by 0.5 s. This fixed time command may reduce the impact of the athlete’s reaction ability on the start [1].

Furthermore, our results were similar to those who reported that the error rate of speed climbing is high. A deeper understanding in the present study is that the error rate in the big and small finals, in which athletes compete for medals, is notably higher than that in the race of other final rounds on average error rate. A previous study showed that after a single climbing training, the strength of sport climbing athletes decreases [20]; in the final stages, the athlete performs one to four high-intensity workouts at high intensity, and the decline in strength can partly explain the increased rate of player errors. In addition, a current study reported that the psychological anxiety of elite athletes in the finals might be more significant than in other competition stages [13]. The level of anxiety may also be responsible for the differences between different final stages.

Additionally, the distribution of error rates in different final stages showed different characteristics with the climbing. The error rate of male athletes in the four phases is relatively average, and the error rate in the middle two phases is higher than in female athletes, while in the P4, female athletes’ error accounted for 46.15% of the total number of female athletes’ errors. Although female climbers experienced significantly more significant adverse effects in comparison to male climbers in speed [1], that does not seem to explain the difference in the distribution of errors between men and women because the total error rate of all athletes in competition is almost the same. The women seem to need relatively longer high-intensity output than men to maintain their climbing speed because, by the time the women had finished the first three phases, the men had already finished the climb. Commonly, explosive power is crucial for elite speed climbers [19]. However, female athletes need to increase their strength output for longer than male athletes to adapt to other characteristics under the current sports level. Thus, in part, different training progress between male and female athletes should be made. Perhaps with the improvement of the explosive power of female athletes, after they reach higher sports levels, the command for explosive endurance may be reduced.

Finally, the error rate of athletes in different body positions is one indicator that has not been studied before. In the stalemate state, the male athlete’s error is 50% higher than that of the female athlete, and the female athlete’s error is twice as significant as that of the male athlete in the trailing state. This may be because the males’ speed racing is more intense than the female, and the climbing speed of males is faster than females. Regarding the difference in the error rate with different body positions, anxiety levels may differ due to the fear of being overtaken or chased and other psychological factors; as found in difficulty climbing, anxiety caused by the fear of falling off can affect a competitor’s performance [13]. Nutrition should also be paid attention to, which is an important factor for athletes to maintain stable performance and reduce the error rate in the multi-round competition system [21,22,23]. To the best of the authors’ knowledge, there are limited findings on athletes’ psychology during speed climbing. Challenging as the monitoring of the psychological changes of athletes in a game is, it may be one of the effective ways to explain the differences in errors.

## 5. Conclusions

In conclusion, the differences between men and women in competition are manifested in many aspects. First, elite male speed climbing is more competitive than females. Second, there was no significant correlation between stages of the route for men’s athletes, while there was a statistical correlation between adjacent stages for women. Third, in terms of errors, the error rate of both men and women in the medal competition stage reached a high rate of ~50%; the error rate of men in each stage of climbing is relatively average, while the error rate of women in the fourth stage accounted for ~46% of the error rate of climbing. However, the psychological and physiological mechanisms that cause this phenomenon have not been thoroughly explored.

## 6. Practical Applications

Notably, the error rate of both males and females is high. Even though both male and female athletes should prioritize the ability to develop the phosphate system and anaerobic power of the muscles, there should be differences in the training progress between genders, the significance of men using the separate phase to training is higher than that of women, and women should pay more attention to the entire route training. Meanwhile, elite female athletes are more capable of high-power output for a longer time than male athletes. Moreover, even after making an error, athletes should accelerate as soon as possible, because speed climbers are more prone to errors in the second half than in the first half. In addition, athletes should train using the official start command consistent with that of IFSC to improve their familiarity with a fixed-time start rather than enhance their reaction ability. We think this strategy also appears to play an essential role in winning a speed competition. We suggest in different final rounds, athletes should choose different climbing strategies based on their opponents’ levels to reduce their error rate.

## Figures and Tables

**Table 1 ijerph-19-06003-t001:** This is a table shows start reaction time, time of each phase and total time.

Phase	Male	Female
Start reaction time	0.20 ± 0.05	0.22 ± 0.06
P1	1.28 ± 0.13	1.66 ± 0.31
P2	1.44 ± 0.16	1.84 ± 0.35
P3	1.15 ± 0.23	1.56 ± 0.27
P4	2.03 ± 0.25	2.66 ± 0.54
Total time	5.90 ± 0.20	7.84 ± 0.43

**Table 2 ijerph-19-06003-t002:** This is a table shows Pearson correlation coefficient between phases in females.

	P1	P2	P3	P4	Total Time
P1	1	0.562 **	0.733 **	−0.028	0.418 **
P2		1	0.823 **	−0.03	0.358 **
P3			1	0.091	0.523 **
P4			0.088	1	0.421 **
Total time					1

** at 0.01 level (two-tailed), the correlation was significant.

**Table 3 ijerph-19-06003-t003:** This is a table shows Pearson correlation coefficient between phases in males.

	P1	P2	P3	P4	Total Time
P1	1	−0.158	−0.132	−0.03	0.355 **
P2		1	−0.211	0.064	0.335 **
P3			1	−0.593 **	0.427 **
P4					0.466 **
Total time					1

** at 0.01 level (two-tailed), the correlation was significant.

**Table 4 ijerph-19-06003-t004:** A comparison between error rates in different competition stages for male and female athletes analyzed in the finals of 6 different competitions.

	Male		Female	
Stage	NTC	ER	NTC	ER
Eighth-finals	96	32.14%	96	32.20%
Quarter-Finals	48	48.21%	48	23.30%
Semi-finals	24	28.57%	24	28.57%
Small finals	12	50.00%	12	57.20%
Big finals	12	57.20%	12	50%
Total	192	35.30%	192	31.30%

(NTC) refers to the total number of climbing attempts seen between all finals stages of competition across 6 different competitions and the error rates (ER) associated with both male and female athletes between all stages.

**Table 5 ijerph-19-06003-t005:** The number of errors in different positions.

Sex	Lead	Stalemate	Lagging	Coefficient of Variation (CV)
Female	24	46	11	49.70%
Male	20	32	20	28.66%

**Table 6 ijerph-19-06003-t006:** Chi-square test of different phases’ error rate in different gender.

Title 1	Phase	NTC	NOEC	NONEC	ER
Female	P1	192	11 a	181 a	5.72%
	P2	185	12 a	173 a	6.49%
	P3	185	15 a	170 a	8.11%
	P4	183	35 b	148 b	19.12%
Male	P1	192	12 a	180 a	6.25%
	P2	186	17 a	167 a	9.14%
	P3	183	18 a	165 a	9.84%
	P4	189	20 c	160 c	11.17%

NTC: Number of total climbs; NOEC: Number of errors climbs; NONEC: Number of no error climbs; ER: Error rate; the same letter after the number indicated no statistical significance between groups, while different letters indicated significant difference with other letter groups (*p* < 0.05).

## Data Availability

Not applicable.

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
