# Peer review of "A Time-Motion and Error Analysis of Speed Climbing in the 2019 IFSC Speed Climbing World Cup Final Rounds"

_ijerph, 2022, doi:10.3390/ijerph19106003_

Round 1
Reviewer 1 Report
Dear Authors,
In my opinion, the article is interesting and properly prepared. It describes the current issues and gives space for further analyzes. I only have a few comments.
I propose to remove figure 1. It doesn't add much.
The full meaning of the abbreviations should be added below the tables.
The summary should add numerical values ​​in the results (e.g. with statistical significance).
It is worth adding the limitations of the study and tips for further ones.
,, the maximum lactic acid value '' it is better to write concentration than value.
,, After the race, the maximum lactic acid value cannot accurately reflect the anaerobic 43
capacity of speed climbers. Moreover, in competitive climbing, anaerobic working ability 44
requirements are different for different disciplines [3]. Thus, speed climbing is a typically 45
short time racing event, and the energy metabolism of the athletes is mainly glycolysis; 46
the explosive power of the athletes is high, so the speed climbing athletes tend to have 47
higher muscle content to meet the requirements of the competition. '' This should be corrected. The substrate for working muscles is not only glucose from glycolysis, but above all, in this case, phosphocreatine and ATP.
There are a few stylistic errors in the work, eg ,, 2.3. procedures '' (lowercase). All text should be checked.
There are very few references in the discussion. It is necessary to compare your results to other scientists (e.g. lines 202-211, 2012-2021 and further).
Kind regards,
Reviewer

Reviewer 2 Report
- There are some grammar/syntax issues that have to be corrected. E.g. line 70-71 - the sentence starting with Twenty seems unfinnished
- The study seems interesting overall, but I do not think that the accuracy is enough as the youtube videos have limited frames/sec; by using these data, the cumulative errors seems significant. Moreover, 50fps is not a standard framerate for youtube videos, but rather 24, 30 or 60.
- statistical analyses are not correctly implemented. For example, I do not see why using Mann-Whitney for a qualitative variable such as gender.
Round 2
Reviewer 1 Report
Dear Authors,
In the future, you may search for information in a bibliography other than through a single entry. For example, the influence of supplementation on climbers was investigated here: Sas-Nowosielski K, Wyciślik J, Kaczka P. Beta-Alanine Supplementation and Sport Climbing Performance. Int J Environ Res Public Health. 2021 May 18; 18 (10): 5370. doi: 10.3390 / ijerph18105370. PMID: 34069981; PMCID: PMC8157844.
In the bibliography of such articles, one can find many references to the characteristics of this discipline. This is a well-known area. The difference is only in the length of the race and a bit of character. In terms of physiology and biochemistry, however, they are still the same human muscles and knowledge about them is predictable and well-known.
Congratulations.
Kind regards,
Reviewer
Author Response
Response to Reviewer 1 Comments
Point 1:
Dear Authors,
In the future, you may search for information in a bibliography other than through a single entry. For example, the influence of supplementation on climbers was investigated here: Sas-Nowosielski K, Wyciślik J, Kaczka P. Beta-Alanine Supplementation and Sport Climbing Performance. Int J Environ Res Public Health. 2021 May 18; 18 (10): 5370. doi: 10.3390 / ijerph18105370. PMID: 34069981; PMCID: PMC8157844.
In the bibliography of such articles, one can find many references to the characteristics of this discipline. This is a well-known area. The difference is only in the length of the race and a bit of character. In terms of physiology and biochemistry, however, they are still the same human muscles and knowledge about them is predictable and well-known.
Congratulations.
Kind regards,
Reviewer
Response 1:
Dear reviewer,
Thank you very much for the comment on improving this study.Based on the comment, we thought about the research conclusion again with other references. We found that nutrition might be one of the essential factors in explaining athletes' ascending stability and added corresponding references in the paper. See lines 259-261 in the revised version, please.
Kind regards,
Authors
Reviewer 2 Report
The authors have improved their article. However, the framerate issue still persists. There is a difference between filming framerate (which depends upon the camera of the videographer) and the streaming framerate (which is wwhat yooutube transmits to the viewer).If the filming and streaming framerates are not similar, error could appear which are important when taking into account the statistical analyses made by the authors
Author Response
Response to Reviewer 2 Comments
Point 1: The authors have improved their article. However, the framerate issue still persists. There is a difference between filming framerate (which depends upon the camera of the videographer) and the streaming framerate (which is wwhat youtube transmits to the viewer).If the filming and streaming framerates are not similar, error could appear which are important when taking into account the statistical analyses made by the authors
Response 1:
Dear reviewer,
Thanks very much for your comments to make this article more scientific. With the past week's efforts, the authors finally found the original video of the 2019 IFSC Speed World Cup. The authors obtained the original video by contacting the video analyst of the China National climbing team who participated in the 2019 World Cup speed Competition with a Sony FDR-AX60 camera, and the video framerate is 100bps. Unfortunately, the Speed climbing World Championship video was not available, so the author deleted the analysis result and expression of the event from the revised manuscript. After five days of effort, we re-collected and determined all the time indicators from the new video. The authors have improved the experimental design, result and relevant sections, please see in the revised version.
Kind regards,
Authors